# The Well-Being of Social Health Professionals: Relationship between Coping Strategies, Emotional Regulation, Metacognition and Quality of Professional Life

**DOI:** 10.3390/ijerph21010051

**Published:** 2023-12-29

**Authors:** Laura Ferro, Marina Cariello, Alessandra Colombesi, Chiara Adduci, Eleonora Centonze, Giorgia Baccini, Stefania Cristofanelli

**Affiliations:** 1Department of Psychology, Faculty of Psychology, University of Valle d’Aosta, 11100 Aosta, Italy; alessandra.colombesi@associazionetiare.org (A.C.); s.cristofanelli@univda.it (S.C.); 2TIARE’, Association for Mental Health, 10125 Turin, Italy; marina.cariello@associazionetiare.org (M.C.); chiara.adduci@associazionetiare.org (C.A.); eleonora.centonze.1995@gmail.com (E.C.); gi.baccini@gmail.com (G.B.)

**Keywords:** social health professionals, quality of professional life, personal resources, coping strategies, emotional regulation, metacognition, minors, burnout, secondary traumatization

## Abstract

Social health professionals should have the knowledge and skills and use personal resources that promote the helping relationship, access to effective intervention strategies, and well-being at work. This study aims to investigate the relationship between some personal resources (coping strategies, emotional regulation and metacognition) and professional satisfaction in a group of social–health professionals working with minors suffering from psychosocial distress. In this professional group, the risk of burnout is common and the quality of professional life is strongly related to the intensity and frequency of exposure to critical and traumatic events. The sample was assessed using self-report instruments: Professional Quality of Life Scale, Coping Orientation to the Problem Experienced, Difficulties in Emotional Regulation Scale and Metacognition Self-Assessment Scale. The quality of professional life showed significant correlations with the psychological characteristics studied. We then tested different regression models: coping orientation scores were found to be a significant predictor of quality of work life for all three components, while emotional dysregulation scores appeared to predict only the burnout component. The quality of professional life of social health professionals was influenced by individual resources at different levels, regardless of knowledge and skills. They showed greater fatigue and aspects of secondary traumatization when emotional disengagement occurred and it seemed to be difficult for them to accept their emotional reactions.

## 1. Introduction

Health and social care professionals are a category of helpers who are in close contact with a target group that is confronted with a variety of stressful conditions. The social health professionals involved in the care of minors may be psychologists, professional educators, nurses and health care workers. The main areas in which these professionals work are residential care, home care and day care in specialized centers. In the Mediterranean context, especially in Spain and Italy, residential care offers a high level of professional qualification, theoretical models and specific working tools that promote and maintain the quality of residential child protection services [1]. In Italy, local administrations define forms and methods of organization and monitoring of residential care for the protection of children and adolescents. This work context places individuals at increased risk of developing work-related stress, as well as increased feelings of fatigue [2,3,4,5]. The role of the health and social worker requires a set of competencies that can be summarized in three categories that include thought (“knowledge”), action (“skills”) and affect (“knowing how to be”). They develop from the combination of knowledge and skills and require the practitioner’s willingness to engage. The terms “knowledge” and “skills” refer to job-specific knowledge, skills and competencies. The “knowing how to be”, on the other hand, refers to the employee’s personal resources that are used in the helping relationship and can influence its quality. This implies a significant emotional and affective investment that enables the helper to understand the needs of the other person and to find appropriate and effective ways to intervene [6].

As in all helping professions, effective intervention occurs through the relational channel, which requires, among other skills, the ability to regulate one’s emotions, select and apply appropriate coping strategies, and use metacognition to evaluate, direct and coordinate all other skills and strategies. Effective social health professionals today must be able to be flexible and adaptive in their activities to the specific situations and needs they encounter in their work practice [7,8]. Moreover, the presence of such skills not only improves employees’ professional competence and, consequently, their ability to interact with users, but also their sense of personal and professional self-efficacy [6]. Caregivers describe how rewarding the role of parenting traumatized children is, even though they may face challenges that impact their well-being [5]. In turn, caregivers recognize the need for support for themselves that is often not available [9]. When employees feel unable to perform their job, they perceive their resources to be insufficient to meet the demands, so they are at risk of burnout [6,10]. Furthermore, those who work with minors within the category of social and health professions face intense emotional, metacognitive and stressful challenges. In fact, the helping relationship between health professionals and children contains specific parenting elements that can significantly impact burnout risk [1]. Furthermore, users may be at risk if caregivers feel stressed or unable to cope with the situation, further exacerbating the child’s difficulties [11,12,13]. Burnout is characterized by emotional exhaustion and lack of job satisfaction and is very common among workers in helping or educational professions [14]. The frequent staff turnover in residential childcare due to burnout, exacerbated by the difficulties in recruiting and retaining staff during COVID-19, has a direct impact on vulnerable children who have already experienced a series of attachment and relationship breakdowns. Staff turnover contributes to the loss of relationships and the unpredictability of children’s lives and is therefore a direct risk factor for these vulnerable children [15]. For these reasons, it is crucial that training interventions for social and health care workers not only target theoretical and/or technical knowledge, but also take into account the skills associated with knowing how to be and serve as a link for all other skills [6,7].

Shealy [16] defined professionals working in these settings as “therapeutic parents”, that is, social health professionals who perform therapeutic activities but have parent-like roles. The professional working with children with psychosocial distress is the primary caregiver for the child and family, assumes direct responsibility for the child and acts as a “surrogate parent” [16,17]. Professionals who have closer daily contact with children as health and social workers are called upon to invest physically and emotionally in children’s physical and mental health to promote their appropriate development and well-being [1,18]. The quality of professional life in emergency services appears to be strongly related to the intensity and frequency of exposure to critical events, since the latter is correlated with increased risk to emergency workers’ mental health [19]. The traumatic experience of a rescuer is not necessarily due to the event itself, but to the helping relationship with a person suffering from the event [20].

Figley [21] proposes the construct of “compassion fatigue”, which describes feelings of deep compassion and understanding for someone affected by suffering, accompanied by a strong desire to alleviate the suffering or eliminate its cause [21]. We can define it as psychological discomfort caused by constant contact with the pain of others, a state of tension and worry characterized by symptomatology reminiscent of post-traumatic stress disorder, which can manifest itself in those who are frequently exposed to the suffering and narrative of the traumatic experiences of others [22]. In this sense, Figley [21] suggests that the response to a critical event lies on a continuum between the positive extreme of job satisfaction (compassion satisfaction) and the negative extreme of attrition (compassion fatigue).

Burnout and compassion fatigue may appear similar, but they differ in several key dimensions. On the one hand, compassion fatigue has an acute and sudden onset that can be triggered by a single experience that the affected person perceives as particularly critical [22]; on the other hand, burnout corresponds to a gradual and progressive exhaustion of the practitioner who feels overwhelmed by his or her work and unable to promote positive change [10,21]. Despite some differences, both can be considered occupational hazards in their own right [23,24,25]. It should be noted that the practice of a helping profession is not necessarily associated with a set of negative consequences. In this context, Stamm [26] introduces the concept of “compassion satisfaction” to describe the positive effects a person may derive from working with suffering people, including positive feelings related to helping others, contributing to the well-being of society, and more generally, the pleasure of “doing one’s job well” [26]. For this reason, the quality of a responder’s professional life cannot be attributed solely to the absence of negative consequences, but to a state of psychosocial well-being, too [27].

It can be stated that the well-being of social health professionals is linked to various aspects and that the quality of the helping relationship requires various relational skills, such as knowing how to regulate one’s emotions, and choosing and using appropriate and utilizing metacognitive skills. If the resources described are perceived as inadequate, professionals are at greater risk of burnout and a reduced quality of professional life, leading to an impoverishment of the helping relationship itself [6,27]. Training would help to improve social health professionals’ awareness and knowledge of disease and increase their ability to understand disease presentations, including the impact and support needed for caregivers. Access to clinical supervision and consultation would enable professionals to reflect on and adapt their care [12,13]. Karadzhov et al. [28] emphasize the importance of asking social health professionals for their reflections on challenges, successes and lessons learned in order to understand the particular needs, struggles and aspirations of these workers in different settings.

In the literature, there are few studies on the quality of life of social health professionals, and most of them are related to patients with chronic diseases [29,30,31] or nurses, particularly after COVID-19 emergencies [32,33,34,35,36].

The aim of the following work was to investigate the relationship between some personal resources (coping strategies, emotion regulation and metacognition) and job satisfaction in a group of social and health professionals working with minors with psychosocial distress. The aim was to use the information obtained not only for the purposes of exploratory research, but also to propose preventive and educational interventions aimed at improving the skills already present in social health professionals and promoting those that are lacking or dysfunctional, as well as to improve the perception of the quality of the actors’ professional life.

## 2. Materials and Methods

### 2.1. Study Design

Data collection was performed by sending a battery of tests through an anonymous Google Form. The collected data were then analyzed using IBM SPSS Statistics version 27 software (SPSS Inc., Chicago, IL, USA). We performed an initial correlation analysis by calculating Spearman’s rank correlation coefficient. Then, different regression models were tested by linear regression analysis.

Ethical approval was obtained from the Ethics Committee of the University of Valle d’Aosta (protocol n. 0015239; 21 September 2023).

### 2.2. Participants

The sample was recruited by sending an email to the coordinators of the residential structures and day-time center for minors throughout Italy. Once consent was received, the test battery was sent. A single institutional email address received all anonymous responses. Therefore, 83 subjects responded to the Google Form, but only 81 had completed the entire battery. They all met our inclusion criteria and therefore made up the sample of our research.

The inclusion criteria for this study were that they were social–health professionals working with minors suffering from psychosocial problems and in need of socio-health care. Only professionals who had been employed for more than three months were included so that the youth relationship had time to establish itself. In particular, a daily report was required, as was typical for those working in residential care centers, in home care services where the professional visits the minor’s home, and in day care where the professionals of a team work in a specialized center that the minor visits during the day and returns home in the evening. We therefore excluded the social health professional figures who do not deal with minors daily, such as structure coordinators, neuropsychiatrists, therapists, social workers, interns and other health professionals responsible for managing internal and external activities at the structure (animal therapists, etc.). Therefore, we included educators, nurses, psychiatric rehab technicians and healthcare assistants.

The mean age was 35.1 years (sd = 10.1), with a mean seniority of 7.8 years (sd = 7.1). The group was composed of 75.3% women (n = 61) and 24.7% men (n = 20). The majority of the sample worked in northern Italy (85.2%) and, in particular, in residential structures (65.4%). More detailed information can be found in Table 1.

### 2.3. Instruments

The test battery consisted of a questionnaire in which some sociodemographic and professional information was requested. This was followed by a battery composed of four self-report questionnaires: Professional Quality of Life Scale (PRQOL), Coping Orientation to the Problems Experienced—New Italian Version (COPE-NVI), Difficulties in Emotional Regulation Scale (DERS) and Metacognition Self-Assessment Scale (MSAS).

Using the PRQOL allows us to understand the quality of professional life of social and health care workers in terms of the risk of developing burnout and compassion fatigue and to assess any elements of satisfaction and gratification that could instead enhance the ability to make personal and organizational efforts that improve the emotional management of their needs and abilities [1,14,37]. Burnout impairs emotional availability and thus therapeutic outcomes for children [38]. Minors who have suffered trauma or other disadvantages need professionals who understand their needs and the emotions underlying the behaviors they perform so that they can learn to self-manage and develop adaptive and resilience skills. This requires the professional to tune into them, show empathy and connect emotionally. These tasks become extremely difficult when one feels exhausted and emotionally exhausted [39].

The emotional regulation scale allows us to understand the ability to identify, recognize and manage one’s emotional experiences. These skills can contribute to “emotional preparedness”, which is able to reduce the negative effects of stressful situations related to the nature of work. The care of minors in social and health professions can be influenced by the impact of individual events that are difficult to deal with, but also by prolonged exposure to situations of suffering, such as previous trauma suffered and/or individual mental health problems of the minors concerned [37]. These working conditions require a high emotional effort, which can lead to the suppression of one’s own emotions and a decrease in sensitivity to traumatic experiences in order to ensure the ability to maintain work duties in the face of great stress. A profession such as that of the participants involved in this research is based on emotions and relationships, and operators must be able to work in this sense to manage and deal with their own emotions and those of the users [37].

For stressful situations to be managed in relation to the minor and the complex care organization in which he/she is incorporated, this could concern the ability to respond adaptively to events and apply useful and effective coping strategies, such as asking for support when necessary, having a positive attitude or planning properly for the implementation of targeted and effective interventions. Our earlier study showed that the use of avoidance strategies appears to be increasing among social and health care workers in the face of greater emotional intolerance [14].

In addition, metacognitive understanding of the other’s psyche could improve the effectiveness of treatment by enabling staff to better understand the minor’s needs [37].

#### 2.3.1. Professional Quality of Life Scale

This questionnaire examines the quality of professional life using 30 items on a Likert scale (from 1 = never to 5 = very often) that explore the experiences in the helping professions [27]. Specifically, the questionnaire identifies two dimensions: compassion satisfaction and compassion fatigue. The first refers to the pleasure the worker derives from being able to do his or her job well. Higher scores on this scale represent greater satisfaction in the ability to be an effective caregiver in one’s job. Compassion fatigue, on the other hand, is the perception of fatigue due to one’s role as a helper in the work context. This dimension is again divided into two sub-dimensions: burnout and secondary traumatic stress. Burnout is known to be a state of stress manifested by feelings of hopelessness and difficulty in coping with work or performing the job effectively. Higher scores on this scale mean you are at higher risk for burnout. Secondary traumatic stress, on the other hand, is the discomfort that results from secondary exposure to extremely or traumatically stressful events. This is a very specific condition of the helping professions and relates to the concept of vicarious traumatization, which is the result of constant exposure to patients’ traumatic experiences. Higher scores do not mean that you have a problem but are an indication that you should change some aspects of your work and/or work environment. The cutoff points refer to each of the three dimensions of the questionnaire and are defined as follows: low for scores of 22 or less; moderate for scores between 23 and 41; and high for scores above 42 [27].

#### 2.3.2. Coping Orientation to the Problems Experienced

The Coping Orientations to Problem Experienced [40] is a self-report questionnaire that considers various coping methods. It includes ratings of five dimensions: social support (seeking understanding and information, and tendency to emotional outbursts), avoidance strategies (denial, use of substances, behavior and mental distancing), positive attitude (attitude of acceptance, containment, and positive reinterpretation of events), problem solving (active and planning strategies) and turning to religion (trust in religion, absence of humor). The Coping Orientation to Problems Experienced—New Italian Version (COPE-NVI) [41] is a 60-item questionnaire with a four-point Likert scale ranging from 1 (I typically do not do it) to 4 (I almost always do it).

#### 2.3.3. Difficulties in Emotional Regulation Scale

This is a self-completion questionnaire consisting of 36 items with Likert responses (from 1 = almost never to 5 = almost always) [42]. The evaluation results in a total score and six subscales. Nonacceptance is the tendency to have a negative secondary response or nonaccepting response to one’s distress; Goals refers to the difficulty in concentrating and/or completing tasks when experiencing negative emotions; Impulse is the difficulty in maintaining control over one’s behavior when experiencing negative emotions; Awareness reflects a lack of awareness or inattention to emotional reactions; Strategies concerns the belief that there is little one can do to self-regulate when upset; Clarity reflects the extent to which a person is aware of and clear about his or her emotions. There are no clinical cutoffs, but high scores indicate greater difficulty in emotional regulation.

#### 2.3.4. Metacognition Self-Assessment Scale

The Metacognition Self-Assessment Scale is a measure of metacognitive skills [43]. This is a self-report measure consisting of 18 items comprising a five-point Likert scale (from 1 = never to 5 = almost always). The assessment includes five dimensions: Monitoring, which represents the ability to identify and name the components of our mental state. Integration, which is the ability to think about different mental states and identify internal contradictions, conflicts and patterns. Differentiation refers to the ability to recognize the representative nature of one’s mental states and mental content. Decentering is the ability to take others’ perspective and hypothesize about their mental states. Finally, mastery refers to the strategies individuals employ to use their knowledge of themselves and others to solve psychological and interpersonal problems. High scores on the MSAS indicate better self-assessment of metacognitive abilities than low scores.

## 3. Results

### 3.1. Descriptive Statistics

Table 2 shows descriptive statistics of the psychological assessment.

As can be seen in Table 2, the mean values of compassion satisfaction and burnout correspond to a moderate level (moderate for scores between 23 and 41), while the mean value of secondary traumatic stress corresponds to a low level (low when the score is 22 or less [27]. We can therefore argue overall that the observed sample of social and health care professionals showed a moderate level of quality of professional life. In particular, from the values summarized in Table 2, it is evident that no subject showed a low level of compassion satisfaction (low when the score is 22 or less), nor a high level of the scale for burnout and secondary traumatic stress (high for scores above 42). With regard to compassion satisfaction, more than half of the sample showed moderate satisfaction (69.1%), while 30.9% reported high satisfaction with the helping profession. Scores were also positive on the secondary traumatic scale, with 71.6% of professionals indicating a low level of discomfort related to secondary trauma. Finally, 70.4% indicated a medium level of burnout, while only a small proportion (29.6%) indicated a low level. However, the three self-administered questionnaires used to measure personal resources do not have cut-offs but are a measure of the psychological characteristics of professionals. COPE values are consistent with normative values [41]. In particular, the sample appears to use social support, positive attitude and problem solving to a slightly greater extent than the normative values. In contrast, the mean values for the use of avoidance strategies and turning to religion are slightly below the normative value. In our sample, therefore, we observe a greater use of proactive strategies.

As for the MSAS and the DERS, a wide range of values is observed, but no minimum or maximum value assumes critical importance. There are no normative values for the MSAS. However, the questionnaire has a total raw score ranging from 18 to 90 [43]. Therefore, we can consider the mean value of the sample as an indicator of an intermediate level of metacognitive abilities. As far as the DERS is concerned, the values given in Table 2 are corrected for age and gender based on the work of Giromini et al. [44]. As these are T-points (mean = 50, SD = 10), it can be argued that the average of our sample corresponds to the normative data [44]. There are greater difficulties with emotional awareness. However, the mean value remains in the range (50 + 10), which can be considered consistent with the intermediate level of metacognitive skills of our sample.

### 3.2. Correlation Analysis

Quality of professional life showed some significant correlations with the psychological characteristics studied. All three components of the PRQOL correlated significantly with some coping strategies and some dimensions of emotional dysregulation. Instead, only secondary traumatic stress appeared to correlate with metacognitive abilities. As can be seen in Table 3, compassion satisfaction correlated positively and with low intensity with positive attitude, problem solving and transcendental orientation.

However, burnout and secondary traumatic stress correlated only moderately with avoidance strategies. Secondary traumatic stress also correlated weakly and directly with social support.

As noted earlier, only secondary traumatic stress appeared to have a correlation, albeit weak, with metacognitive abilities. Specifically, a direct correlation is observed with the MSAS monitoring component and an indirect correlation with the integration component. Finally, emotional dysregulation correlated with low and moderate intensity with all components of professional quality of life. The highest correlation was observed between secondary traumatic stress and the DERS goals dimension. Stress resulting from secondary traumatic stress increased with difficulty concentrating and/or accomplishing tasks when experiencing negative emotions. This dimension of the PRQOL correlated positively with all DERS aspects except the clarity component.

Compassion satisfaction correlated weakly and inversely with the DERS components of nonacceptance, strategies and the total scale. Burnout showed the same correlations, this time directly. A correlation with DERS_goals was also observed.

### 3.3. Regression Analysis

We performed linear regression analyses using the dimensions measured by PRQOL as dependent variables. We then tested several regression models using the COPE, MSAS and DERS dimensions as independent variables. The MSAS demonstrated no significance with any of the three PRQOL dimensions. Otherwise, COPE proved to be a significant predictor of professional quality of life, for all three components. DERS appears to predict only the burnout component (Figure 1).

Starting from these observations, we conducted a regression analysis with PRQOL_Compassion Satisfaction as the dependent variable and all components of COPE as the independent variable (R^2^ 0.27; SE = 4.46; *p* = 0.00). Thus, the coping strategies measured with the COPE were able to explain 27% of the compassion satisfaction scores measured with the PRQOL. Table 4 summarizes the coefficients of each predictor. COPE_problem solving (B = 0.27; SE = 0.14; *p* = 0.05) and COPE_turning to religion (B = 0.45; SE = 0.18; *p* = 0.01) were significant predictors.

We therefore entered PRQOL_burnout as the dependent variable and COPE and DERS as independent variables (R^2^ = 0.34; SE = 4.57; *p* = 0.00). The two dimensions together could explain 34% of the burnout values. The significant predictors were COPE_avoidance strategies (B = 0.64; SE = 0.20; *p* = 0.00) and DERS_non-acceptance (B = 0.17; SE = 0.06; *p* = 0.00) (Table 5).

Finally, we tested the model with PRQOL_secondary traumatic stress as the dependent variable and COPE as the independent variable (R^2^= 0.24; SE = 4.69; *p* = 0.00). An amount of 24% of the secondary traumatic stress values could be explained by the values measured in COPE. As shown in Table 6, COPE_social support (B = 0.18; SE = 0.09; *p* = 0.05) and COPE_avoidance strategies (B = 0.71; SE = 0.19; *p* = 0.00) were significant predictors for this dimension.

The results of the regression analyses thus indicate the predictive role of coping strategies on professional quality of life. Burnout, one of the three dimensions of professional quality of life, also appears to be partly influenced by the ability to regulate emotions. Metacognitive skills, on the other hand, show no predictive role for the three PRQOL dimensions investigated.

## 4. Discussion

The purpose of this study was to examine the relationship between personal resources such as coping strategies, metacognitive and emotional regulation skills, and quality of professional life within a specific occupational group characterized by aspects of caregiving.

Impairment of the quality of professional life and the associated risk of developing burnout syndrome are more common among workers who interact with others and are involved in professional relationships such as helping, supporting and guiding [45,46]. Professionals who work with minors are significantly affected by aspects of burnout that could compromise the quality of their professional lives, precisely because of the emotional commitment that caring for minors requires [1,47]. Healthcare professionals tend to underestimate and fail to manage their state of stress, leading to a worsening of their situation [16,19]; burnout affects professionals’ emotional availability and therapeutic outcomes in minors [39]. When emotional exhaustion is present, it becomes difficult to tune into the caregiving relationship and be empathetic [21,25]. In addition, burnout conditions can cause frequent turnover, which negatively impacts children who have already experienced relationship breakdowns and losses [15]. In addition, it appears that under these conditions, there is a tendency to become emotionally detached, to use avoidance strategies and to reduce the ability to regulate emotions [14].

Specific aspects of the quality of professional life were examined with the PROQOL [27], namely compassion satisfaction, burnout and secondary traumatic stress. None of the three dimensions was impaired in our sample. Our sample also had moderate scores on the personal resources examined, such as emotional regulation skills, metacognition and appropriate use of coping strategies.

All dimensions of professional quality of life are related to coping strategies and some aspects of emotional dysregulation. Notably, however, professional satisfaction seems to depend exclusively on coping strategies implemented at cognitive and behavioral levels, such as transcendental orientation and problem orientation.

Thus, the hypothesis is that the enjoyment of the helping relationship with minors is influenced by the perception of being acting and engaged subjects, i.e., by the perception of feeling actively involved in their intervention with concrete actions and being able to bring about positive change through operational strategies and planning.

Moreover, personal satisfaction, although positively associated with some aspects of emotional regulation, does not seem to depend on these aspects. At the same time, it is completely independent of metacognitive abilities. Therefore, it is possible that it is related to a dimension of utility that is more likely to be achieved through the use of cognitive and behavioral strategies that may be more timely than through recourse to the other skills studied.

The dimension of fatigue and especially secondary traumatization is instead associated with higher levels of emotional dysregulation and reduced metacognitive abilities. Fatigue in the helping relationship also appears to depend on the tendency toward disengagement that results from resorting to the avoidance and nonacceptance of responses to emotional distress. Consistent with the previous study [14], it can be hypothesized that this tendency may limit the use of appropriate coping strategies, emotional regulation and metacognitive skills, which the sample has been shown to possess and which could be a protective dimension in this case.

In addition, it is possible that the tendency toward avoidance could contribute to a picture of emotional dysregulation associated with fatigue by reducing opportunities to process and cope with negative experiences. These findings seem relevant because in the helping relationship with minors, the healthcare professional is the one who simultaneously interacts with his or her own emotions and influences the other [48]. In educational practice, the educator must constantly demonstrate their ability to listen and be in an empathetic position, becoming the “container” through their words, gestures and actions, and being the one who attributes meaning and significance to the emotional experiences of the minors, which are often unverbalized and therefore are usually enacted. These emotional and communication skills must then be translated into purposeful and deliberate actions that educate and repair. The educator’s position is precisely that of a “participating witness” who constantly observes and participates even in stressful conditions associated with the rhythms of everyday life [49].

## 5. Conclusions

This study extends the findings of previous research on social and health care professionals involved in the care of minors with psychosocial problems. After the pandemic, they showed a significant deterioration in their emotional skills as well as avoidance and emotional suppression strategies, which were strongly associated with a state of emotional exhaustion and an associated risk of burnout [14].

This professional group has the peculiarity of defining itself as a health profession that simultaneously performs a care and support function. The literature shows that the professions most likely to develop burnout symptoms are precisely those in which relationships are helpful and supportive.

The quality of professional life of social and healthcare workers who work with minors suffering from psychosocial problems is influenced at different levels by individual resources, regardless of knowledge and skills. This study was also conducted during a period following the COVID-19 health emergency, which significantly altered the lives of professionals working in care and support relationships, increasing perceptions of stress and specific burnout symptoms such as emotional exhaustion, emotional distancing and depersonalization, as well as a decrease in feelings of satisfaction and fulfilment. Among health care professionals caring for minors, the pandemic required the use of numerous emotional and cognitive resources. In particular, given the greater emotional exhaustion and lower satisfaction with their work, the increase in avoidance strategies was evident. These findings also emerged in the present study, which confirms that these professionals experience greater fatigue and aspects of secondary traumatization when emotional disengagement occurs and the difficulty of accepting their own emotional responses appears.

Therefore, investing in these resources, such as appropriately strengthening coping strategies and improving the ability to use metacognitive and emotional resources, can lead to greater satisfaction and prevent the risk of burnout and secondary traumatization. The quality of the work of the professionals, a good organizational network that provides adequate support and regular supervision that allows the processing of emotional experiences related to the profession are aspects recognized as protective factors for the well-being of the operators and a better quality of intervention that promotes the well-being of the minors involved [28,39].

Organizations operating in this field could promote awareness and understanding of the skills associated with “knowing how to be”, as well as through digital tools that can provide useful elements for the mental health of professionals remotely and with flexible access and can promote social support. Supervisors could also provide models for the effective use of strategies to manage one’s own well-being and potentially stressful situations. Supervisors should focus on professionals’ emotional reactions to professional challenges and encourage them to process and reflect on them, as well as pay attention to their health and interpersonal skills [28].

Moreover, a better ability to recognize and communicate one’s emotions could facilitate communication with colleagues with whom one shares the work with minors, with a positive impact on collaboration and the effectiveness of the intervention; a good team relationship makes it possible to resolve complaints and conflicts, including those related to users’ pathologies, which could have an impact on the functioning of the team itself [50].

Appropriate spaces for reflection and sharing can modulate the maladaptive use of avoidance and promote a relationship with the minor in which care develops through adult and appropriate mental functioning, emotional experiences can be processed and managed, and intervention and action can be more consciously oriented [51].

## Figures and Tables

**Figure 1 ijerph-21-00051-f001:**
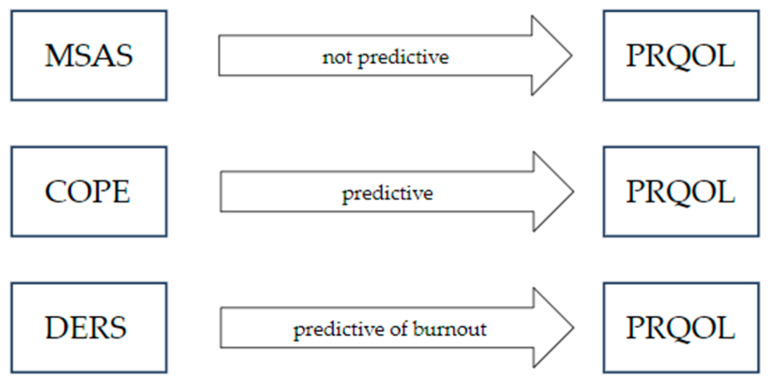
Linear regression model.

**Table 1 ijerph-21-00051-t001:** Respondent profile: sociodemographic characteristics.

Characteristics	Category	Percentage	Mean (SD)
Gender	Female	75.3%	
	Male	24.7%	
Age			35.1 (10.1)
Educational Level	Bachelor’s degree	39.5%	
	Master’s degree	7.4%	
	Specialization	24.7%	
	Professional qualification	28.4%	
Site	Centre	2.5%	
	North	85.2%	
	South	12.3%	
Marital status	No partner	27.2%	
	Married	25.9%	
	Divorced	7.4%	
	Partner cohabiting	29.6%	
	Partner no cohabiting	9.9%	
Workplace	Day-time center	18.5%	
	Residential structure	65.4%	
	Independent profes.	1.2%	
Minors in care (numbers)	1–3	2.5	
	4–7	11.1%	
	8–10	60.5%	
	11+	25.9%	

n = 81.

**Table 2 ijerph-21-00051-t002:** Psychological assessment.

	Minimum Value	Maximum Value	Mean Value	Stand. Deviation Value
PRQOL_compassion satisfaction	24	49	38.72	5.053
PRQOL_burnout	15	39	25.80	5.221
PROQOL_secondary traumatic stress	11	36	19.73	5.211
COPE_social support	16	46	30.93	6.292
COPE_avoidance strategies	16	31	20.63	2.943
COPE_positive attitude	20	45	33.98	5.486
COPE_problem solving	23	44	33.27	4.881
COPE_turning to religion	10	28	19.05	3.290
MSAS_monitoring	15	30	24.06	3.075
MSAS_integration	4	10	7.62	1.410
MSAS_differentation	5	10	7.98	1.351
MSAS_decentering	8	15	12.65	1.825
MSAS_mastery	14	25	19.11	2.434
MSAS_total	46	74	60.30	6.075
DERS_NonAcceptance	36.11	86.65	48.34	9.706
DERS_Goals	29.17	69.85	45.597	7.404
DERS_Impulse	35.98	62.26	46.120	6.229
DERS_Awarness	33.65	87.98	60.014	13.996
DERS_Strategies	36.44	69.65	46.436	6.793
DERS_Clarity	35.20	63.10	52.030	6.793
DERS_Total	29.88	67.88	49.258	8.053

n = 81.

**Table 3 ijerph-21-00051-t003:** Correlation analysis.

		PRQOL_Compassion Satisfaction	PRQOL_ Burnout	PRQOL_Secondary Traumatic Stress
COPE_social support	corr. coeff.	0.095	0.144	0.256 *
	sign.	0.400	0.201	0.021
COPE_avoidance strategies	corr. coeff.	−0.211	0.409 **	0.446 **
	sign.	0.059	0.000	0.000
COPE_positive attitude	corr. coeff.	0.371 **	−0.051	−0.046
	sign.	0.001	0.653	0.684
COPE_problem solving	corr. coeff.	0.356 **	0.037	−0.047
	sign.	0.001	0.754	0.678
COPE_turning to religion	corr. coeff.	0.320 **	0.039	0–098
	sign.	0.004	0.727	0.386
MSAS_monitoring	corr. coeff.	−0.206	0.115	0.242 *
	sign.	0.065	0.307	0.029
MSAS_integration	corr. coeff.	0.127	−0.096	−0.295 **
	sign.	0.257	0.394	0.008
MSAS_differentiation	corr. coeff.	−0.019	0.073	0.082
	sign.	0.863	0.518	0.465
MSAS_decentering	corr. coeff.	−0.007	−0.127	−0.080
	sign.	0.948	0.259	0.479
MSAS_mastery	corr. coeff.	0.214	0.041	−0.105
	sign.	0.055	0.716	0.352
MSAS_total	corr. coeff.	0.119	−0.079	−0.189
	sign.	0.290	0.485	0.091
DERS_NonAcceptance	corr. coeff.	−0.226 **	0.431 **	0.323 **
	sign.	0.043	0.000	0.003
DERS_Goals	corr. coeff.	−0.205	0.289 **	0.478 **
	sign.	0.067	0.009	0.000
DERS_Impulse	corr. coeff.	−0.024	0.207	0.393 **
	sign.	0.832	0.063	0.000
DERS_Awarness	corr. coeff.	−0.195	0.081	−0.003
	sign.	0.082	0.472	0.980
DERS_Strategies	corr. coeff.	−0.328 **	0.268 *	0.436 **
	sign.	0.003	0.016	0.000
DERS_Clarity	corr. coeff.	−0.169	−0.010	0.022
	sign.	0.132	0.929	0.845
DERS_Total	corr. coeff.	−0.288 **	0.293 **	0.395 **
	sign.	0.009	0.008	0.000

n = 81. ** The correlation is significant at the 0.01 level (two-tailed). * The correlation is significant at the 0.05 level (two-tailed).

**Table 4 ijerph-21-00051-t004:** Regression analysis. PRQOL_Compassion Satisfaction is dependent variable, COPE is independent variable.

	B	SE	Standardized B	t	Sign.
(Constant)	25.518	6.347		4.021	0.000
COPE_social support	−0.124	0.088	−0.154	−1.412	0.162
COPE_avoidance strategies	−0.195	0.181	−0.114	−1.076	0.285
COPE_positive attitude	0.101	0.120	0.109	0.836	0.406
COPE_problem solving	0.272	0.138	0.263	1.974	0.052
COPE_turning to religion	0.450	0.177	0.293	2.545	0.013

**Table 5 ijerph-21-00051-t005:** Regression analysis. PRQOL_Burnout is dependent variable, COPE and DERS are independent variables.

	B	SE	Standardized B	t	Sign.
(Constant)	3.048	8.670		0.352	0.726
COPE_social support	0.040	0.094	0.048	0.429	0.669
COPE_avoidance strategies	0.637	0.203	0.359	3.137	0.003
COPE_positive attitude	0.009	0.130	0.010	0.072	0.943
COPE_problem solving	−0.045	0.148	−0.042	−0.301	0.765
COPE_turning to religion	0.148	0.188	0.093	0.789	0.433
DERS_NonAcceptance	0.173	0.064	0.323	2.702	0.009
DERS_Goals	0.135	0.096	0.192	1.416	0.161
DERS_Impulse	−0.003	0.125	−0.004	−0.026	0.980
DERS_Awarness	−0.004	0.050	−0.012	−0.089	0.929
DERS_Strategies	−0.041	0.137	−0.053	−0.301	0.765
DERS_Clarity	−0.109	0.118	−0.141	−0.920	0.361

**Table 6 ijerph-21-00051-t006:** Regression analysis. PRQOL_Secondary Traumatic Stress is dependent variable, COPE is independent variable.

	B	SE	Standardized B	t	Sign.
(Constant)	−3.968	6.679		−0.594	0.554
COPE_social support	0.181	0.092	0.219	1.961	0.054
COPE_avoidance strategies	0.711	0.191	0.402	3.731	0.000
COPE_positive attitude	−0.055	0.127	−0.057	−0.431	0.668
COPE_problem solving	−0.020	0.145	−0.019	−0.137	0.891
COPE_turning to religion	0.312	0.186	0.197	1.677	0.098

## Data Availability

The datasets that were generated for this study are available upon request from the corresponding author.

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
