# Peer review of "The Well-Being of Social Health Professionals: Relationship between Coping Strategies, Emotional Regulation, Metacognition and Quality of Professional Life"

_ijerph, 2023, doi:10.3390/ijerph21010051_

Round 1
Reviewer 1 Report
Comments and Suggestions for Authors Dear authors, you should review the quality of the citations and increase their number in the discussion, those used are of little relevance and the number is insufficient. You should also review the bibliography, updating it to citations less than 5 years old, which exist on the subject and would give greater quality to your scientific text.Author Response
uploaded file

Reviewer 2 Report
Comments and Suggestions for Authors
In the article you write that:
The study is part of a broader work aimed at analyzing the professional well-being 305 of socio-educational professionals working with minors who present a psychosocial risk.
What kind of research project is this, describe it in more detail in the section: Participants
More information on ethical criteria.
Author Response
uploaded file

Reviewer 3 Report
Comments and Suggestions for Authors
This manuscript provides a relatively large contribution to the literature on the Well-Being of Social-Health Professionals, in relation with coping strategies, emotion regulation, and quality of life.
With this being said, there are a few areas where minor revisions and elaborations are needed before this manuscript is ready for publication. Therefore, I suggest a minor revision on the language, together with a more elaborated discussion on the implication and impact of the study for health professionals.
Comments on the Quality of English LanguageI kindly advice minor revisions and proofreads on the quality of English
Author Response
uploaded file

Reviewer 4 Report
Comments and Suggestions for Authors
Review: the well-being of social health professionals: relationship between coping strategies…
This paper talks about social health professionals but this is a term I have not come across before. I assume they are some sort of counsellor. The paper looks at the well-being of this group regarding their personal resources and satisfaction.
Abstract
just a minor point I’m not used to seeing acronyms presented in the abstract. Perhaps you could write generally about the scales used
Introduction
Now the authors talk about helping professionals and then health and social workers. Late,r on page 1, I read about social-educational professionals. The authors really need to choose a term and stick with it.
Yes, this type of work is stressful and tiring and people doing this work need to regulate their emotions, have coping strategies and see the big picture. Avoiding burnout is important and this should be covered in training.
Working with high-risk children will be especially challenging and may lead to compassion fatigue. At line 77 you say compassion fatigue is sudden and acute: I would like to see some academic support for that statement. At line 82; what is an auxiliary occupation? At line 90 you say “there are few studies of quality-of-life” really??
Materials and Methods
81 people completed a series of psychological tests online. How were they accessed? Why 81? You need to justify why this was a number thought suitable for your sample. At 2.3 line 119 what is “anamnestic”?
You then describe the four scales. I think we need a bit more background about why these scales were important to use in this study and link these scales with literature. At the top of page 5 there is a table without any labelling
Results
What is a good level of quality of professional life? I feel that the information about the cut-off points should really be given earlier in the description of the scales. We need more commentary about the other scales results.
3.2 correlation analysis
coping and professional quality-of-life appear to be related significantly
3.3
the results of the regression analysis need careful explaining. You have just dropped them in there without any commentary. Would a visual framework perhaps be helpful? I found it difficult to understand this section. It does seem that disengagement occurred in some cases. Please spell out these results meaningfully in a way that people can grasp.
Discussion
it seems that your sample did okay on the professional quality-of-life test and on the other scales. Professional quality-of-life is related to coping and apparently some aspects of emotional dysregulation. I didn’t understand your comment at line 275 about being acting and active subjects.
Conclusion
What is the broader work that this is part of? Yes, no doubt the period following the pandemic presented extra stresses for these people.
In general
I was surprised that there didn’t appear to be any ethics approval.
Please attend to the length of your paragraphs: there is a huge paragraph on page 2 which needs to be broken down.
Comments on the Quality of English LanguageThe English language here is not too bad
Author Response
uploaded file

Round 2
Reviewer 4 Report
Comments and Suggestions for Authors
Second review: the well-being of social health professionals….
I’m pleased to see that my suggestions about the abstract of been adopted.
In the introduction, there is additional information about who our health and social care professionals as requested.
Happy to see information included about compassion fatigue, as well as other information and references requested about this topic.
Regarding the materials and methods, there is considerable additional information about how recruiting respondents was conducted. I’m also pleased to see significant additional information about the scales used in the study.
Results: once again additional explanatory material has been included
Discussion: the section has been slightly boosted in good way.
Conclusion: the ethics approval statement has been included earlier in the paper
overall
the authors have included my requested alterations in a very satisfactory way.